# Circ-ERC2 Is Involved in Melatonin Synthesis by Regulating the miR-125a-5p/MAT2A Axis

**DOI:** 10.3390/ijms232415477

**Published:** 2022-12-07

**Authors:** Hai-Xiang Guo, Yi Zheng, Guo-Kun Zhao, Hao-Qi Wang, Song Yu, Fei Gao, Jia-Bao Zhang, Yong-Hong Zhang, Bao Yuan

**Affiliations:** Department of Laboratory Animals, College of Animal Sciences, Jilin University, Changchun 130062, China

**Keywords:** rat, pineal, melatonin, SAM, MAT2A, ncRNA, synthesis

## Abstract

The circadian rhythm of melatonin secretion in the pineal gland is highly conserved in vertebrates. Melatonin levels are always elevated at night. Acetylserotonin O-methyltransferase (ASMT) is the last enzyme in the regulation of melatonin biosynthesis (N-acetyl-5-hydroxytryptamine-melatonin). S-adenosylmethionine (SAM) is an important methyl donor in mammals and can be used as a substrate for the synthesis of melatonin. Methionine adenosyltransferase (MAT) catalyzes the synthesis of SAM from methionine and ATP and has a circadian rhythm. CircRNA is an emerging type of endogenous noncoding RNA with a closed loop. Whether circRNAs in the pineal gland can participate in the regulation of melatonin synthesis by binding miRNAs to target mat2a as part of the circadian rhythm is still unclear. In this study, we predicted the targeting relationship of differentially expressed circRNAs, miRNAs and mRNAs based on the results of rat pineal RNA sequencing. Mat2a siRNA transfection confirmed that mat2a is involved in the synthesis of melatonin. Circ-ERC2 and miR-125a-5p were screened out by software prediction, dual-luciferase reporter experiments, cell transfection, etc. Finally, we constructed a rat superior cervical ganglionectomy model (SCGx), and the results showed that circ-ERC2 could participate in the synthesis of melatonin through the miR-125a-5p/MAT2A axis. The results of the study revealed that circ-ERC2 can act as a molecular sponge of miR-125a-5p to regulate the synthesis of melatonin in the pineal gland by targeting mat2a. This experiment provides a basis for research on the circadian rhythm of noncoding RNA on pineal melatonin secretion.

## 1. Introduction

The pineal gland is an important neuroendocrine organ in the body, and its main function is to secrete melatonin under the control of the hypothalamus [1]. In lower vertebrates, the pineal gland has direct photosensitivity; however, in mammals, including humans, the pineal gland has lost direct photosensitivity [2,3]. Rodents are often used as experimental animal models and are widely used in the study of regulating pineal melatonin secretion [4,5,6]. Melatonin plays a role in a variety of physiological functions in the body, and melatonin deficiency may lead to the impairment of liver and mitochondrial function in nonalcoholic fatty liver disease (NAFLD) [7]. Evidence indicates that melatonin and its analogs play an important role in insomnia and depression [8]. The circadian rhythm of melatonin secretion is controlled distally by the suprachiasmatic nucleus (SCN) and then regulated proximally in response to norepinephrine (NE) [9]. The synthesis of melatonin requires the participation of four enzymes, namely tryptophan hydroxylase (TPH), aromatic amino acid decarboxylase (AADC), serotonin N-acetyltransferase (AANAT) and hydroxyindole-O-methyltransferase (HIOMT) [10]. AANAT is significantly differentially expressed in the pineal gland during the day and night, and it is a restriction enzyme in the process of melatonin synthesis; most of the related studies choose AANAT as an important research object [11,12]. HIOMT is the enzyme that ultimately synthesizes melatonin, and it does not have a diurnal differential expression pattern similar to melatonin and AANAT [11]. Studies have shown that AANAT is not the rate-limiting enzyme of melatonin biosynthesis, but the rhythm-producing enzyme of melatonin synthesis [9,13]. Moreover, a missense mutation of HIOMT was found in patients with autism spectrum disorder, resulting in a significant decrease in the secretion of melatonin, which confirmed the key role of HIOMT [14]. Although the diurnal difference in HIOMT in the pineal gland is not significant, the relationship between the reaction rate and the substrate of the last step of melatonin synthesis is unknown.

S-adenosylmethionine (SAM) is an important methyl donor in mammals and an essential substrate for the final step of melatonin synthesis [15,16]. SAM is differentially expressed diurnally in the pineal gland; that is, it exhibits increased secretion during the day and decreased secretion at night, and in contrast to the expression pattern of melatonin, a study showed that SAM is heavily consumed at night as a methyl donor for melatonin synthesis [16,17]. S-adenosylmethionine is synthesized by methionine adenosyltransferase (MAT) catalyzed by methionine and ATP [18,19]. MAT exists in three forms, namely MATⅠ, MATⅡ, and MATⅢ. Among them, mat1a encodes MATⅠ and MATⅢ, which are only expressed in the adult liver, and mat2a encodes MATⅡ protein, which is mainly expressed in the pineal gland [17,20,21]. The microarray results of the pineal gland indicate that mat2a is differentially expressed day and night in the rat pineal gland and is associated with melatonin synthesis [22]. In addition, studies have shown that mat2a is similar to aanat in the rat pineal gland and is directly regulated by NE-cAMP in the pineal gland [17]. Our previous findings also revealed that mat2a was significantly upregulated at night in the rat pineal gland [23]. However, there is still a lack of key evidence regarding whether MAT is involved in the synthesis and secretion of melatonin.

Posttranscriptional regulatory mechanisms play an increasingly important role in circadian rhythm regulation [24,25]. CircRNA is a covalently closed circular molecule in eukaryotic cells that can act as a molecular sponge for miRNA to regulate gene expression post-transcriptionally. Studies have shown that circRNAs are highly expressed in the rat pineal gland and exhibit a circadian rhythm [23]. Among them, circR-WNK2 has been shown to act as a molecular sponge for miR-328a-3p and promote the expression of AANAT in the rat pineal gland [26]. In the rat pineal gland, miR-483 has been reported to target aanat to regulate melatonin synthesis [27]. In the zebrafish pineal gland, miR-183 was also found to bind to the mRNA 3′-UTR of NFIL3-6 and AANAT2 to reduce melatonin synthesis [28]. Studies have found that miR-451 can reduce the protein expression level of 14-3-3, which may be related to low melatonin levels in patients with autism spectrum disorder [25]. Recent studies have found that miR-325 can target the 3′-UTR of AANAT mRNA in both humans and rats, which may be related to the decreased level of melatonin in neonatal ischemic brain injury [29,30]. The ceRNA network composed of circRNAs, miRNAs, and mRNAs in the rat pineal gland may play a key role in the circadian rhythm and synthesis of the pineal gland [23].

In this study, to prove that mat2a is involved in the synthesis of melatonin, we demonstrated the correlation between mat2a and melatonin by detecting the expression changes of mat2a after circadian or NE treatment. The regulatory relationship between the two was verified by mat2a siRNA treatment. Differentially expressed circRNAs and miRNAs were screened based on RNA-seq results. The role of post-transcriptional regulatory mechanisms in melatonin synthesis was explored through cell transfection, dual-luciferase reporter assays, and fluorescence in situ hybridization experiments. Our study explores the regulatory relationship between the circ-ERC2/miR-125a-5p/MAT2A axis and melatonin synthesis, complements the regulatory mechanism of rat melatonin synthesis, and provides a reference for mammalian circadian rhythm research.

## 2. Results

### 2.1. Correlation between Mat2a and Melatonin Synthesis

Presequencing results indicate that mat2a is differentially expressed in the pineal gland of diurnal rats [23]. Mat2a shares a similar expression pattern with AANAT and melatonin, namely, low expression during the day and high expression at night. The expression levels of mat2a and aanat in daytime and nighttime rat pineal glands were detected, and it was found that the expression levels of mat2a and aanat were significantly upregulated at night, but there was no significant difference in the expression of asmt (Figure 1A). The protein expression changes of AANAT, ASMT and MAT2A were consistent with the gene expression changes (Figure 1B). ELISA results showed that melatonin expression levels were significantly upregulated at night, and SAM was significantly downregulated at night (Figure 1C). We then performed immunofluorescence colocalization of MAT2A, ASMT, and AANAT in the pineal gland, and the results showed that the three proteins were in the same cell type in the pineal gland (Figure 1D).

We also performed NE treatment experiments, and RT‒qPCR results showed that the expression levels of mat2a, asmt and aanat were significantly increased after NE stimulation. The protein expression of MAT2A was also significantly increased, while the protein expression level of ASMT showed no significant difference. Changes in the expression of melatonin and SAM after NE treatment were consistent with circadian changes, and related results after NE treatment can be found in the Appendix A.

### 2.2. Mat2a Is Involved in the Synthesis of Melatonin

To explore the regulatory relationship between mat2a and melatonin synthesis, we used mat2a siRNA treatment in rat pineal cells. Among them, siRNA-1 had the best effect, so siRNA-1 was used for subsequent experiments (Figure 2A). The expression of mat2a was knocked down in pineal cells with and without NE treatment. The RT‒qPCR results showed that the expression level of mat2a was significantly downregulated in both groups after siRNA-1 treatment, and the overall expression level of the NE-treated group was significantly upregulated (Figure 2B). The expression of MAT2A protein in rat pineal cells was also significantly decreased after Mat2a siRNA transfection (Figure 2C). The ELISA results showed that the expression level of melatonin was significantly decreased, and the expression level of SAM was significantly increased (Figure 2D). However, the levels of melatonin synthesis-related genes, including aanat, asmt, and tph1, did not change significantly (Figure 2E).

### 2.3. MiR-125a-5p Participates in Melatonin Synthesis by Regulating the Expression of mat2a

Based on previous rat pineal RNA sequencing results, we used the TargetScan, miRanda, and RNAhybrid software to predict the targeting relationship between circadian differentially expressed miRNAs and mat2a in the sequencing results. The predicted results are shown in Appendix A. In this study, we screened these mat2a-targeting miRNAs. According to the results, there was no significant change in the expression level of mat2a after transfection of miR-185-5p, miR-30c-5p and miR-351-5p mimic in rat pineal cells (data unpublished). The expression level of mat2a was significantly downregulated in rat pineal cells after transfection of miR-125a-5p, miR-125b-1-3p and miR-103-3p mimic (data unpublished). Since miR-125b-1-3p was less expressed in the pineal gland, the luciferase activity of wild-type mat2a was not changed after miR-103-3p transfection (Appendix A).

In this study, miR-125a-5p was finally selected as a candidate miRNA for subsequent experiments. RT‒qPCR results demonstrated that miR-125a-5p was expressed at low levels in the rat pineal gland at night, and the expression of miR-125a-5p was significantly downregulated compared with that during the daytime (Figure 3A). The targeting relationship between miR-125a-5p and mat2a was verified by a dual-luciferase reporter assay, and it was found that the luciferase activity of wild-type mat2a was significantly downregulated after transfection with the miR-125a-5p mimic (Figure 3B). However, the luciferase activity was not significantly upregulated after mutation of the three predicted targeting sites (Appendix A). Using miR-125a-5p mimic and inhibitor to transfect rat pineal cells, it was found that the expression level of mat2a was significantly downregulated in the transfected mimic group, and the expression level of mat2a was significantly upregulated in the transfected inhibitor group (Figure 3C). We also detected MAT2A protein expression changes after miR-125a-5p mimic/inhibitor transfection, and the results were consistent with the RT‒qPCR results (Figure 3D). Finally, we detected the expression of miR-125a-5p in different tissues of daytime rats. The results showed that miR-125a-5p was expressed in various tissues of rats and was significantly enriched in brain tissues and organs (Figure 3E).

### 2.4. Identification of Circ-ERC2

We used TargetScan, miRanda and RNAhybrid to predict the targeting relationship between the differentially expressed circRNAs and miR-125a-5p in the sequencing results, and the prediction results are shown in Appendix A. According to the sequence of circ-ERC2 in the sequencing results, the structure of circ-ERC2 was drawn (Figure 4A). To confirm the circular structure of circ-ERC2, we designed a divergent primer for circ-ERC2 and sequenced the PCR product. The sequencing results of circ-ERC2 spanning splice sites were consistent with the sequence alignment of circ-ERC2 (Figure 4B). Random and oligo DT-specific reverse transcription primers were used for fluorescence quantification, and the expression of circ-ERC2 in the oligo DT reverse transcription group was significantly downregulated compared with that in the random group (Figure 4C). In the linear RNase digestion experiment, there was no obvious change before and after linear RNase digestion in the divergent primer group, while the convergent primer group showed a significant downregulation after linear RNase digestion (Figure 4D). RT‒qPCR detected the expression of circ-ERC2 in different tissues of rats during the day and found that circ-ERC2 was significantly enriched in brain organs and lungs (Figure 4E). Finally, we performed fluorescence in situ hybridization (FISH) on circ-ERC2 in rat pineal cells. The FISH results showed that circ-ERC2 was mainly distributed in the cytoplasm of rat pineal cells (Figure 4F).

### 2.5. Circ-ERC2 sIs Involved in Melatonin Synthesis via the miR-125a-5p/mat2a Axis

To verify the sequencing results, we detected the expression changes of circ-ERC2 in the pineal gland of rats during the day and night and found that circ-ERC2 was highly expressed at night (Figure 5A). Dual-luciferase reporter experiments to verify the targeting relationship between circ-ERC2 and miR-125a-5p revealed that the luciferase activity of wild-type circ-ERC2 was significantly downregulated after transfection of miR-125a-5p, but it did not recover after mutating the predicted target site (Figure 5B and Appendix A). Circ-ERC2 siRNA was transfected into rat pineal cells, and siRNA-1 with relatively stable experimental results was selected for subsequent experiments (Figure 5C). To verify whether circ-ERC2 can affect the expression of downstream mat2a and miR-125a-5p, knockdown experiments were performed. The expression level of mat2a was significantly decreased after transfection with circ-ERC2 siRNA, while the expression level of miR-125a-5p was significantly increased (Figure 5D). The protein expression level of MAT2A was also significantly decreased (Figure 5E).

We co-transfected circ-ERC2 siRNA and miR-125a-5p inhibitor into rat pineal cells and found that miR-125a-5p inhibitor could rescue the significant downregulation of mat2a due to circ-ERC2 knockdown (Figure 5F). The changes in MAT2A protein expression were consistent with those in the RT‒qPCR results (Figure 5G).

### 2.6. SCG Regulates Melatonin Synthesis via the circ-ERC2/miR-125a-5p/mat2a Axis

The study detected the expression changes of circ-ERC2 and miR-125a-5p in rat pineal cells treated with NE in vitro. The results showed that the expression level of miR-125a-5p was decreased after NE treatment, while the expression level of circ-ERC2 was increased, but the difference was not significant (Figure 6A). Since the expression of some genes in the pineal gland is regulated by the hypothalamus-SCG axis, we detected the expression changes of noncoding RNAs in the pineal gland after removing the SCG in rats. The results showed that aanat, mat2a, and circ-ERC2 in the pineal gland of the control group were significantly upregulated at night, and the diurnal difference was lost after SCG removal. The expression level of miR-125a-5p in the pineal gland of the control group was significantly decreased at night, and the circadian difference was also lost after SCG removal (Figure 6B).

## 3. Discussion

The synthesis of melatonin in the mammalian pineal gland requires the participation of various proteins such as TPH1 [31], ASMT [32], and AANAT [33]. Among them, AANAT has been shown to have diurnal differential expression and is regulated by NE-cAMP [34]. In this study, the differential expression change of AANAT was used as a control, which could reflect the regulation of the pineal gland in response to NE. We found that the mRNA and protein expression levels of AANAT were differentially expressed between day and night, and NE treatment also significantly increased the expression level of AANAT. For ASMT, there is evidence that ASMT expression levels show only small diurnal differences or no apparent circadian rhythm, and measurements of ASMT activity in the pineal gland also show only very small diurnal differences [11,32,35]. Our results also revealed that the expression level of ASMT was only slightly or not significantly increased at night and after NE treatment. The current study determined that MAT2A is highly expressed in the rat pineal gland and is regulated by the NE-cAMP pathway [17,36]. The results of rat microarray analysis classified mat2a into melatonin synthesis genes [22]. Rat single-cell sequencing results found that mat2a was significantly upregulated at night in β-pineal cells [5]. Our findings also found that MAT2A was differentially expressed during the day and night in the rat pineal gland and was regulated by NE. However, there are not enough studies to prove the effect of mat2a expression changes on melatonin synthesis.

We examined the expression levels of melatonin and SAM in diurnal rat pineal glands and NE-treated cells. The results showed that the expression level of melatonin was significantly upregulated at night, and the expression level of SAM was significantly downregulated, which was consistent with the results of other studies [16]. To verify whether the changes in nocturnal SAM were regulated by SCG, we added NE to rat pineal cells to simulate the nocturnal environment of the rat pineal gland. The results of the study revealed that NE treatment could significantly increase the secretion of melatonin in serum and simultaneously decrease the content of SAM in serum. Studies have shown that the expression level of melatonin in the systemic circulation of eight weeks old rats is in the range of 50–70 pg/mL [37]. The expression level of melatonin in rat serum in this study was also in this range. However, due to the high sensitivity of ELISA, the use of ELISA to detect the changes of hormone expression in rat serum has certain limitations. Moreover, we used FISH to determine the localization of AANAT, ASMT and MAT2A in rat pineal tissue. The results showed that the three proteins had similar positions in the pineal gland, which further indicated that MAT2A may work together with AANAT and ASMT to participate in the synthesis of melatonin in the rat pineal gland. Based on circadian rhythm studies, AANAT was proposed to be the rate-limiting enzyme in melatonin synthesis [13]. To explore whether MAT2A expression changes can affect the synthesis of melatonin, we transfected mat2a siRNA into rat pineal cells in this study. The results showed that there was no significant change in melatonin synthesis-related genes, while the expression level of melatonin was significantly decreased. Although transfection of mat2a siRNA cannot completely eliminate the expression of mat2a, this method can be used to specifically knock down mat2a in vitro and has certain advantages. We also detected the expression levels of melatonin and SAM after transfection with mat2a siRNA and found that the expression level of melatonin was significantly decreased, and the secretion of SAM was significantly increased. This may be due to the decreased efficiency of melatonin synthesis and the decreased rate of SAM consumption, leading to the accumulation of SAM in pineal cells. These results suggest that the expression of mat2a affects melatonin synthesis.

Research on post-transcriptional regulation in melatonin synthesis has gradually expanded, and studies have shown that miR-451 [25], miR-483 [27], miR-325 [29], miR-7 [38] and other miRNAs are involved in the synthesis of melatonin. Previous studies in our laboratory revealed that circRNA-WNK2 could act as a ceRNA to regulate the expression of AANAT in the rat pineal gland [26]. Studies have shown that a variety of miRNAs, including miR-101-5p [39], miR-136 [40], miR-942-5p [41], and miR-485-5p [42], can target the MAT2A mRNA 3’-UTR, thus regulating cancer progression. However, the expression of these miRNAs in the rat pineal gland showed no significant diurnal variation. This study is the first to report the regulatory effect of miRNAs on MAT2A in the rat pineal gland. In this study, a variety of differentially expressed circRNAs were found based on the RNA sequencing results of day and night rat pineal glands [23]. Since circRNAs and mRNAs contain multiple miRNA binding sites (MREs), the TargetScan, miRanda, and RNAhybrid software were used to predict their targeting relationships and build a ceRNA regulatory network. In the previous results, six differentially expressed miRNAs were screened during the day and night, and the expression level of mat2a was significantly decreased after transfection of three miRNA mimics. Due to the low expression level of miR-125b-1-3p in the rat pineal gland, follow-up related studies could not be carried out. The expression level of mat2a was significantly decreased after miR-103-3p mimic treatment, but the dual luciferase reporter assay showed that there was no targeting relationship between the two. The results indicated that miR-103-3p may regulate the expression of mat2a through other pathways. Therefore, in this study, miR-125a-5p and circ-ERC2 were screened out for follow-up research. Dual-luciferase reporter experiments verified the targeting relationship between miR-125a-5p and mat2a and circ-ERC2 and miR-125a-5p. However, after we mutated all the sites predicted by the software, there was no obvious recovery of luciferase activity. This may be due to the limitations of the prediction software, resulting in not being able to find all binding sites. Current evidence suggests that circ-ERC2 can sponge miR-125a-5p to regulate mat2a expression.

We also noticed that there were significant differences in gene expression in rat pineal cells treated with NE, but they failed to reach the level of diurnal expression. For example, the circadian difference of AANAT in the rat pineal body is as high as approximately 150 times [43]. In this study, the expression level of AANAT only increased by approximately 70 times after NE treatment. At the same time, we found that although the expression trend of circ-ERC2 and miR-125a-5p existed after NE treatment, there was no significant difference, which may be due to the difference in NE treatment concentration in vitro and in vivo. To verify whether circ-Ecr/miR-125a-5p/MAT2A is regulated by SCG and the accuracy of the in vitro experimental results, we constructed a rat superior cervical ganglionectomy model according to the procedure [44]. This model has been widely used in the rat pineal gland. Malena L. Mul Fedele et al. verified through experiments that SCGx is a good model for studying the effects of the biological clock on neuroendocrine function. In addition, superior cervical ganglionectomy (SCGx) can be used in melatonin application studies to reduce the effects of endogenous melatonin [45]. Ptosis of the upper eyelid of the rat was used as the criterion for us to determine the success of the model [44], and the pineal gland of the rat was taken for the follow-up study one week later. The RT‒qPCR results showed that the differences in mat2a, miR-125a-5p and circRNA disappeared, and the changes before and after AANAT surgery were also used as indicators of the success of the model in this study. How the superior cervical ganglion (SCG) regulates the circ-ERC2/miR-125a-5p/MAT2A axis remains to be studied. Finally, based on the findings of this study, we mapped the mechanism by which the circ-ERC2/miR-125a-5p/mat2a axis regulates melatonin synthesis under the action of the superior cervical ganglion (Figure 7).

In conclusion, this study determined that MAT2A plays an important role in the process of melatonin synthesis and that MAT participates in the process of melatonin by promoting the synthesis of SAM. The mechanism by which circ-ERC2 participates in melatonin synthesis in the rat pineal gland through the miR-125a-5p/MAT2a axis was also discovered, which provides a reference for exploring the circadian rhythm in mammals.

## 4. Materials and Methods

### 4.1. Ethics Statement

Our experimental procedures were carried out in strict accordance with the Guidelines for the Care and Use of Laboratory Animals of Jilin University. Healthy eight weeks old male SD rats were provided by Liaoning Changsheng Co., Ltd. (Liaoning Province, Shenyang, China). and all experimental protocols were approved by the Institutional Animal Care and Use Committee of Jilin University (license number: SY202212031). Animal experiments were performed at the Experimental Animal Center of Jilin University, and animals were euthanized using a carbon dioxide anesthesia machine.

The rats were exposed to light at 12:12 (light: night) during the feeding process. Daytime pineal sampling was usually performed 6 h after the onset of light, and nighttime pineal sampling was performed 6 h after the onset of darkness. Light intensity (150 lx).

### 4.2. Pineal Cell Culture and Processing

To ensure that the experimental procedure remained sterile, we used autoclaved ophthalmic scissors and ophthalmic forceps to remove the rat head and dissociated the rat head skin, and the pineal gland was removed after opening the skull. The tissues were washed in prechilled phosphate buffered saline (BI, Shanghai, China), and all manipulations were performed near an alcohol lamp.

Ophthalmic forceps were used to dissect the pineal gland from the nerve to which was attached, and then the intact pineal gland was transferred into prechilled PBS with 0.3% bovine serum albumin (BSA) and 1% penicillin/streptomycin (BI, Shanghai, China). Collagenase type I (Soleibao, Beijing, China) and DMEM (BI, Shanghai, China) were used to prepare a 2.5% collagenase solution, which can digest 4–6 pineal glands per ml of collagenase solution. The pineal gland tissue was ground into pieces using a sterile 1 mL syringe, and then the pineal gland was placed in a 37 °C, 5% CO_2_ incubator for 30 min digestion, and the tissue was completely digested by repeated pipetting. Then, a 70-µm cell sieve was used for filtration, the filtrate was collected, fetal bovine serum (FBS) (BI, Shanghai, China) was added to stop digestion, and the sample was centrifuged at 200× *g* for 10 min at room temperature. After centrifugation, the supernatant was carefully discarded and the cell pellet was diluted with 10% fetal bovine serum (FBS) DMEM. The diluted cell suspension was transferred to a 12-well plate, one well for every two pineal glands. Finally, the twelve-well plate was placed in a 37 °C, 5% temperature-controlled incubator for cell culture and monitoring of cell status. Unless otherwise specified, NE was added to rat pineal cells at a concentration of 3 µM, and the treatment time was 6 h [46].

### 4.3. Transfection with mat2a siRNA, miR-125a-5p-mimic, miR-125a-5p-inhibitor, circ-ERC2 siRNA

Rat pineal cells were cultured in a constant temperature incubator at 37 °C for four days before transfection, during which the medium was changed every two days. The siRNA, mimic and inhibitor used in the transfection process were all from Ribo (Guangzhou, China). The transfection process was carried out in strict accordance with the protocol recommended by Ribo. For RT‒qPCR, the transfection concentration and transfection time of mat2a-siRNA were 100 nM and 24 h. The transfection concentration and transfection time of miR-125a-5p-mimic were 100 nM and 48 h, respectively. The transfection concentration and transfection time of miR-125a-5p-inhibitor were 100 nM and 24 h, respectively. The transfection concentration and transfection time of circRNA-ERC2 siRNA were 100 nM and 24 h, respectively. For Western blotting, the transfection time of each group was increased by 24 h.

### 4.4. RNA Isolation and RT‒qPCR

The rat pineal tissue collection procedure was the same as above. Both tissue and cellular RNA were extracted using the TRIzol (Yamei, Shanghai, China) extraction method. First-strand cDNA was synthesized using Monad’s (Wuhan, China) MonScript™ RTIII ALL-in-One Mix with a dsDNase kit. The first-strand cDNA of miR-125a-5p was synthesized using the FastKing RT kit from Tiangen (Beijing, China). RT‒qPCR was performed using the MonAmp™ ChemoHS qPCR Mix kit from Monad (Wuhan, China). All primer sequences used during RT‒qPCR can be found in Appendix A. Among them, GAPDH was used as the internal control gene for mRNA and circ-ERC2, and U6 was used as the internal control gene for miR-125a-5p.

### 4.5. ELISA Detection of Melatonin and SAM Expression Levels

The pineal cell supernatant was stored at 4 °C, and the rat blood was stored at 4 °C overnight. The samples were centrifuged at 1000× *g* for 20 min to remove impurities and cell debris, and then the supernatant was collected for detection. The rat melatonin ELISA kit was from Enzyme-linked Biotechnology Co., Ltd. (Shanghai, China). The kit was removed from the refrigerator 60 min in advance and allowed to equilibrate to room temperature (18–25 °C). Standard wells, blank wells and sample wells were set, and 50 µL standard/sample dilution/sample was added to be tested. Then, 50 µL of biotinylated antibody working solution was added to the sample wells and incubated at 37 °C for 30 min after covering. Then, we used washing solution to wash each well, added 350 µL washing solution to each well, washed 5 times in total, and soaked for 1 min at a time. Except for the blank wells, 100 µL of horseradish peroxidase (HRP)-labeled detection antibody was added to each well, incubated at 37 °C for 30 min after covering, and washed five times. Then, 50 µL of each substrate solution (A and B) was added and incubated at 37 °C for 15 min after covering. Finally, 50 µL of stop solution was added to stop the reaction, and the optical density (OD) value of each well at a wavelength of 450 nm was detected immediately by a microplate reader.

For Melatonin ELISA kit, the sensitivity of ELISA kit is 9.38 pg/mL, and the detection range is 15.63–1000 pg/mL. It can detect the expression level of melatonin in rats in the samples, and there is no obvious cross-reaction with other analogues.

### 4.6. Prediction of ceRNA Regulatory Networks

In this experiment, based on the RNA sequencing results of the rat pineal gland in the early stage of the laboratory, circRNAs and mRNAs with high expression at night and miRNAs with high expression during the day were screened out. The selection process used the TargetScan, Miranda and RNAhybrid native software. The intersection was used to construct the rat pineal ceRNA regulatory network.

### 4.7. Dual-Luciferase Reporter Assay

The dual-luciferase reporter results in this experiment were obtained by Ribo (Guangzhou, China). The pmiR-RB-REPORT™ dual luciferase reporter vector was used, and the reporter fluorescence of the vector was the Renilla luciferase gene (hRluc). Fluorescence was calibrated using the firefly luciferase gene (hluc) as an internal reference. The 3’UTR of Mat2a or circ-ERC2 was cloned downstream of the hRluc gene. Since miR-125a-5p has a targeting relationship with target genes, miR-125a-5p and the constructed vector were co-transfected into 293T cells, and the relative fluorescence value of the reporter gene was downregulated to verify the interaction between miRNA and target genes.

### 4.8. Western Blotting

Adherent cell protein extraction was performed using RIPA lysis buffer from Yamei (Shanghai, China). After discarding the medium, the cells were washed with 1×PBS, and 120 µL lysis buffer and 1.2 µL protease inhibitor were added. Pipette evenly so that the lysate is in full contact with the cells. After placing the six-well plate in a 4 °C refrigerator for 30 min, all the liquid was transferred to a new centrifuge tube. The samples were centrifuged at 16,000× *g* for 10 min at 4 °C, and the supernatant was carefully transferred to a new centrifuge tube. The Biyuntian (Shanghai, China) BCA Protein Concentration Assay Kit (Enhanced) was used to detect the protein concentration. A microplate reader was used to measure the absorbance value of the sample at a wavelength of 562 nm, and the standard concentration of the sample was calculated according to the standard curve. Proteins were denatured using 4× protein loading buffer from Solarbio (Beijing, China) at 95 °C for 10 min.

Gels were prepared using the PAGE Gel Quick Preparation Kit of Yamei (Shanghai, China), which should be equilibrated to room temperature before use. The two-color pre-stained protein marker was purchased from Yamei (Shanghai, China). The electrophoresis conditions were 120 V and 120 min, and the electroporation conditions were 220 mA and 50 min. BCA protein was dissolved in 1× TBST solution to prepare blocking solution, blocked at 37 °C for 1 h, and then incubated with the antibody overnight. GAPDH antibody was from Cell Signaling Technology (CST) (1:1000, Shanghai, China, 2118S), MAT2A + MAT1A antibody was from Abcam (1:1000, ab177484), AANAT antibody was from Abcam (1:1000, ab3505), and ASMT antibody was from GeneTex (1:1000, GTX130292). The membrane was then washed three times with 1× TBST for 10 min each. The goat anti-rabbit antibody was obtained from Cell Signaling Technology (CST) (1:4000, Shanghai, China, 7074S). After incubating the secondary antibody for 1 h at room temperature on a shaker, the above washing steps were repeated. Finally, development was performed using the Omni-ECL™ Ultra-Sensitive Chemiluminescence Detection Kit (femtogram grade) from Yamei (Shanghai, China).

For the specificity of the antibodies, GAPDH (14C10) Rabbit mAb #2118, Cell Signal-ing Technology, and the antibody was validated by the company using confocal immuno-fluorescence and flow cytometry, please refer to the manufacture’s description: https://www.cellsignal.cn/products/primary-antibodies/gapdh-14c10-rabbit-mab/2118?site-search-type=Products&N=4294956287&Ntt=2118s&fromPage=plp&_requestid=6017262, accessed on 2 December 2022. MAT2A + MAT1A antibody [EPR10496] ab177484, Abcam, and the antibody was validated by the company using confocal immunofluorescence and flow cytometry, please refer to the manufacture’s description: https://www.abcam.cn/nav/primary-antibodies/rabbit-monoclonal-antibodies/mat2a--mat1a-antibody-epr10496-ab177484.html, accessed on 2 December 2022. AANAT antibody (ab 3505), Abcam, and the anti-body was validated by a study, please refer to the description: https://www.ncbi.nlm.nih.gov/pmc/articles/PMC6152527/, accessed on 2 December 2022. ASMT antibody, GTX130292, GeneTex, and the antibody was validated by the company by Western blot. please refer to the manufacture’s description: https://www.genetex.cn/Product/Detail/ASMT-antibody/GTX130292#datasheet, accessed on 2 December 2022.

### 4.9. Sanger Sequencing and Linear RNase Treatment

CircRNA-ERC2 target sites were amplified using 2 × Taq Plus PCR MasterMix from Tiangen (Beijing, China). The amplification procedure was pre-denaturation at 94 °C for 3 min, followed by 40 cycles of 30 s at 94 °C, 30 s at 60 °C, and 20 s at 72 °C, and finally a total extension at 72 °C for 5 min. The final PCR product was sequenced by Kumei (Changchun, China). The process of identifying circRNAs by linear RNase treatment was the same as in previous studies in the laboratory [47].

### 4.10. Fluorescence In Situ Hybridization (FISH)

Fam-labeled circ-ERC2 probes were designed and synthesized by Suzhou Gene Pharmaceutical Co., Ltd. (Suzhou, China). A fluorescence in situ hybridization kit was used to determine the location of circ-ERC2, and the hybridization procedure was performed according to the manufacturer’s instructions. The specific operation steps can refer tobe found in our previous study [26]. Images were finally captured on a fluorescence microscope (Nikon).

For pineal tissue section immunofluorescence, AANAT, ASMT, MAT2A primary antibodies and Western blotting were the same. Among them, FITC-labeled fluorescent secondary antibody was used for AANAT, Cy5-labeled fluorescent secondary antibody was used for ASMT, and CY3-labeled fluorescent secondary antibody was used for MAT2A.

### 4.11. Construction of a Rat Model of Superior Cervical Ganglionectomy

The rat superior cervical ganglionectomy model was established in strict accordance with the procedure [44]. The SD rats were divided into two groups: the experimental group underwent surgery on both sides of the superior cervical ganglia, and the control group underwent surgery on the same site that did not damage the ganglia. After the operation, the rats were reared individually under 12:12 light and dark conditions, and the state of the rats was observed. Rats were euthanized with carbon dioxide at the indicated time points one week later, and the pineal glands were harvested for subsequent experiments.

### 4.12. Statistical Analysis

All experiments in this study were repeated at least 3 times. The t test of GraphPad Prism 8 was used to compare the significance of the two groups of data, and the multiple comparison data were analyzed for significance using SPSS 19.0 one-way ANOVA. All data are the mean ± standard deviation of three independent biological replicates. *p* < 0.05 was considered to indicate statistical significance.

## 5. Conclusions

In conclusion, this study determined that MAT2A plays an important role in the process of melatonin synthesis and that MAT participates in the process of melatonin by promoting the synthesis of SAM. The mechanism by which circ-ERC2 participates in melatonin synthesis in the rat pineal gland through the miR-125a-5p/MAT2a axis was also discovered, which provides a reference for exploring the circadian rhythm in mam-mals.

## Figures and Tables

**Figure 1 ijms-23-15477-f001:**
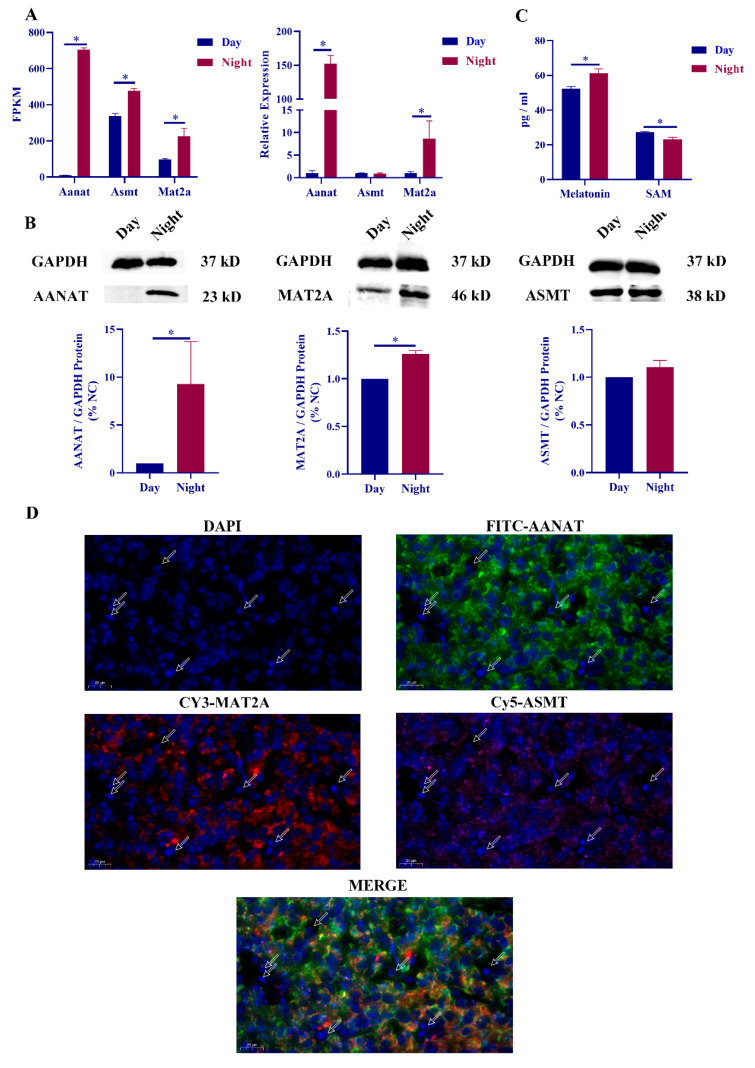
Mat2a was co-expressed with melatonin synthesis gene. (**A**) Relative expression of aanat, asmt and mat2a mRNA in RNA-seq and RT-qPCR results during the day and night. (**B**) Relative expression of AANAT, ASMT and MAT2A proteins in pineal gland during the day and night. (**C**) Relative expression of melatonin and S-adenosine methionine (SAM) in rat serum during the day and night. (**D**) Immunofluorescence colocalization of AANAT (green), ASMT (pink) and MAT2A (red) in rat pineal gland. Arrows indicate areas of low gene expression with a scale of 20 µm. *, *p* < 0.05.

**Figure 2 ijms-23-15477-f002:**
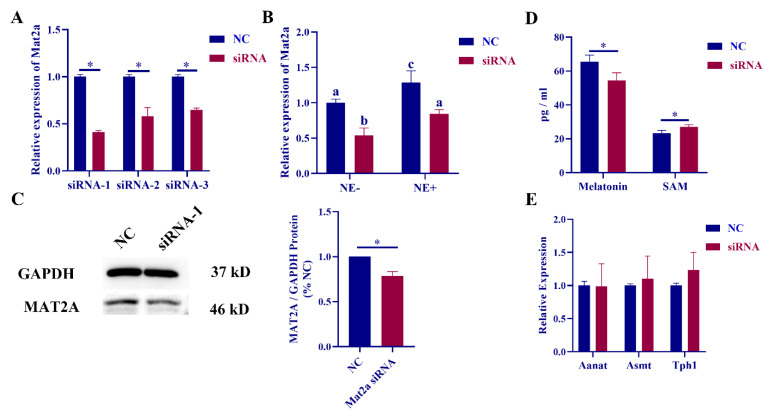
Interference with mat2a expression inhibits melatonin synthesis. (**A**) Relative expression of mat2a mRNA in rat pineal cells transfected with mat2a siRNA. (**B**) Relative expression of mat2a mRNA after co-treatment with NE and mat2a siRNA-1. The letters (a, b, c) mean there are significant differences among them. (**C**) Relative expression of MAT2A protein expression after mat2a siRNA-1 transfection. (**D**) Relative expression of melatonin and S-adenosine methionine (SAM) content in pineal cell supernatant after transfection with mat2a siRNA-1. (**E**) Relative mRNA expression of melatonin synthesis-related genes after mat2a siRNA-1 transfection. *, *p* < 0.05. NC means Mat2a siRNA-NC.

**Figure 3 ijms-23-15477-f003:**
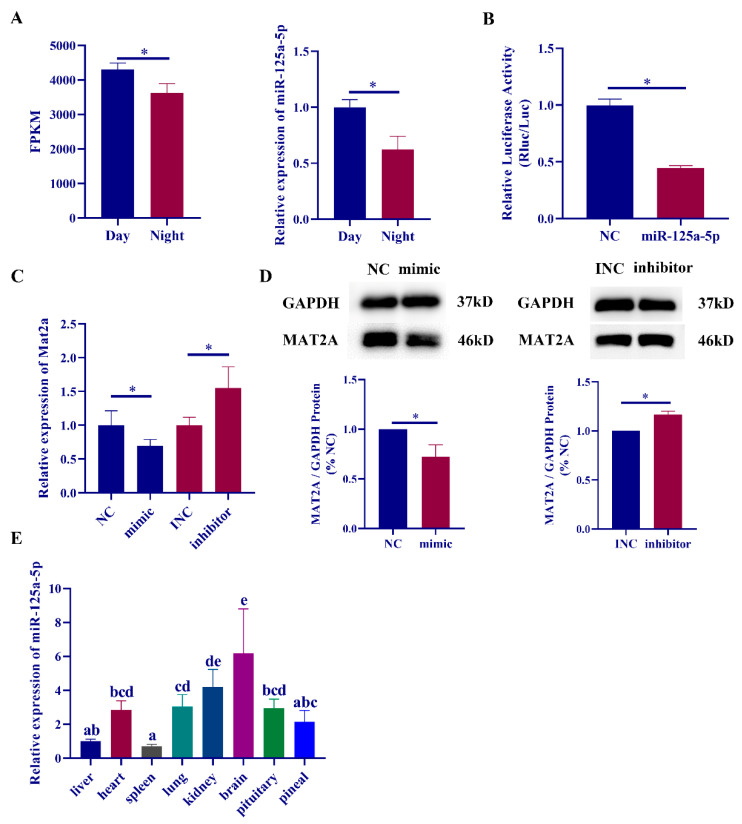
miR-125a-5p binds to mat2a mRNA 3′-UTR and inhibits MAT2A expression. (**A**) Relative expression of miR-125a-5p in the pineal gland of rats during the day and night. (**B**) Dual luciferase reporter assay results for miR-328a-3p, the wild-type mat2a mRNA 3′-UTR. (**C**) Relative expression of mat2a mRNA after transfection with miR-125a-5p mimic/inhibitor. (**D**) Relative expression of MAT2A protein after transfection with miR-125a-5p mimic/inhibitor. (**E**) Relative expression of miR-125a-5p in different tissues of rats. The letters (a, b, c, d, e) mean there are significant differences among them. *, *p* < 0.05. NC means miR-125a-5p mimic-NC and INC means miR-125a-5p inhibitor-NC.

**Figure 4 ijms-23-15477-f004:**
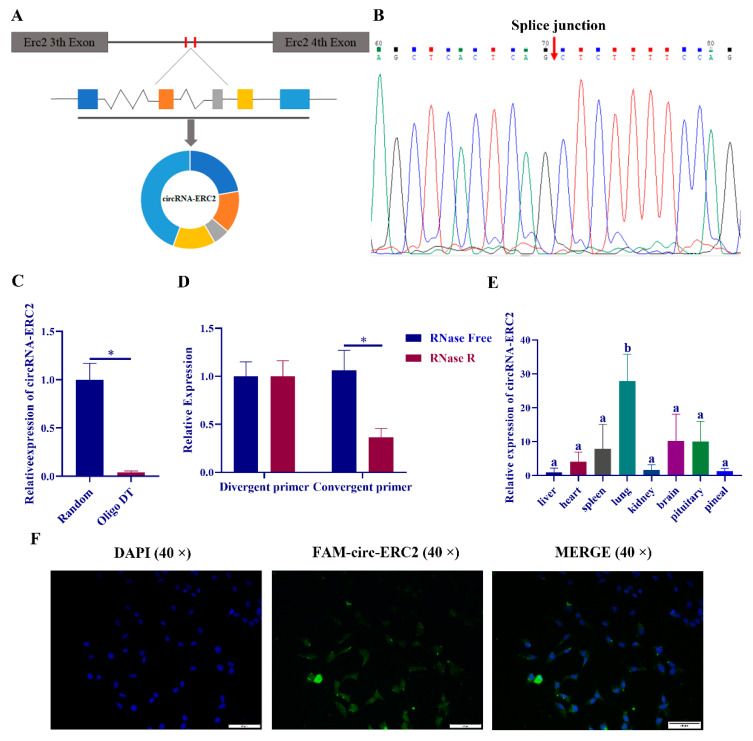
Structural identification and cellular localization of circ-ERC2 in rat pineal gland. (**A**) Chromosomal localization of circR-ERC2. (**B**) Identification of the circR-ERC2 sequence. (**C**) Relative expression of circR-ERC2 with oligo (dT) or random reverse transcription primers. (**D**) Expression of circ-ERC2 and linear-ERC2 before and after linear RNase treatment. (**E**) Relative expression of circ-ERC2 in different tissues of rats. The letters (a, b) mean there are significant differences among them. (**F**) Representative images of FISH show the localization of circR-WNK2 (green). Nuclei were counterstained with DAPI (blue). *, *p* < 0.05.

**Figure 5 ijms-23-15477-f005:**
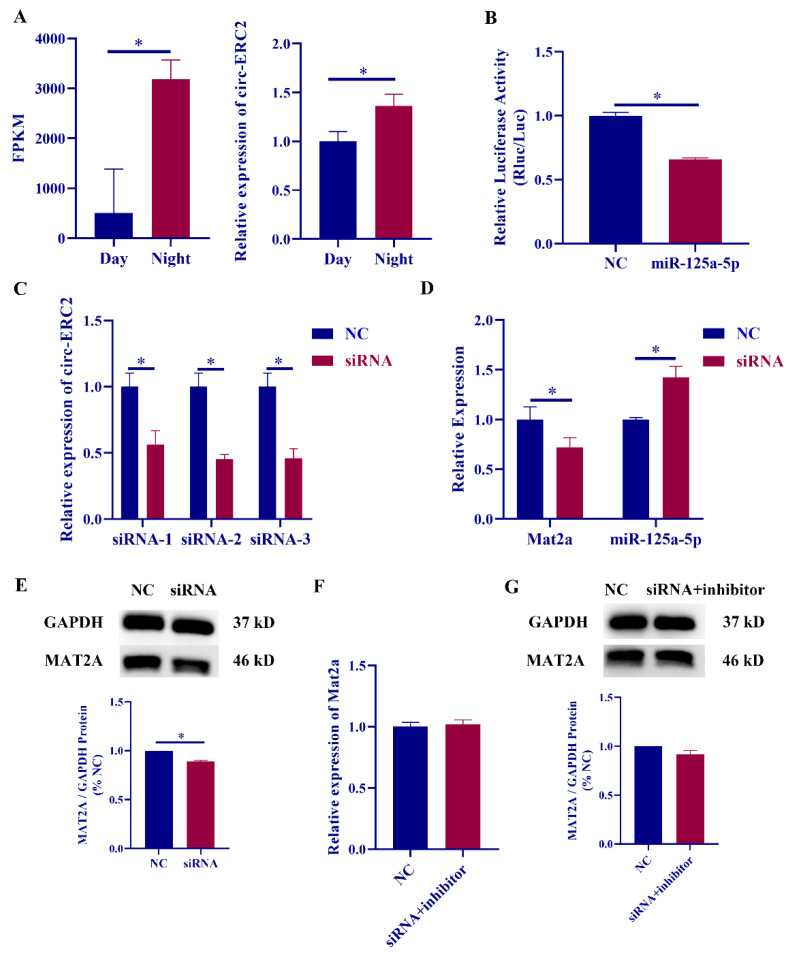
Circ-ERC2 acts as a molecular sponge for miR-125a-5p in pineal cells. (**A**) Relative expression of circ-ERC2 in rat pineal gland during the day and night. (**B**) Dual luciferase reporter assay results for miR-328a-3p, the wild-type circ-ERC2. (**C**) Relative expression of circ-ERC2 in rat pineal cells transfected with circ-ERC2 siRNA. (**D**) Relative expression of mat2a mRNA and miR-125a-5p after circ-ERC2 siRNA-1 transfection. (**E**) Relative expression of MAT2A protein after circ-ERC2 siRNA-1 transfection. (**F**) Relative expression of mat2a mRNA after circ-ERC2 siRNA-1 and miR-125a-5p inhibitor co-transfection. (**G**) Relative expression of MAT2A protein after circ-ERC2 siRNA-1 and miR-125a-5p inhibitor co-transfection. *, *p* < 0.05.

**Figure 6 ijms-23-15477-f006:**
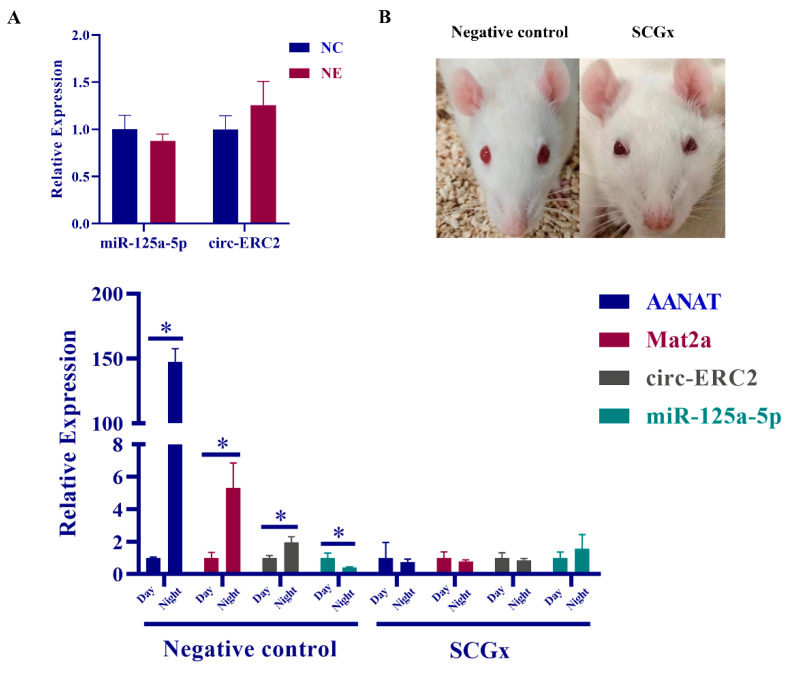
Superior cervical ganglion (SCG) can participate in melatonin synthesis through circ-ERC2/miR-125a-5p/MAT2A axis. (**A**) Relative expression of miR-125a-5p and circ-ERC2 in pineal cells before and after NE treatment. (**B**) Relative expression of genes in the pineal gland of control and SCGx groups during the day and night. *, *p* < 0.05.

**Figure 7 ijms-23-15477-f007:**
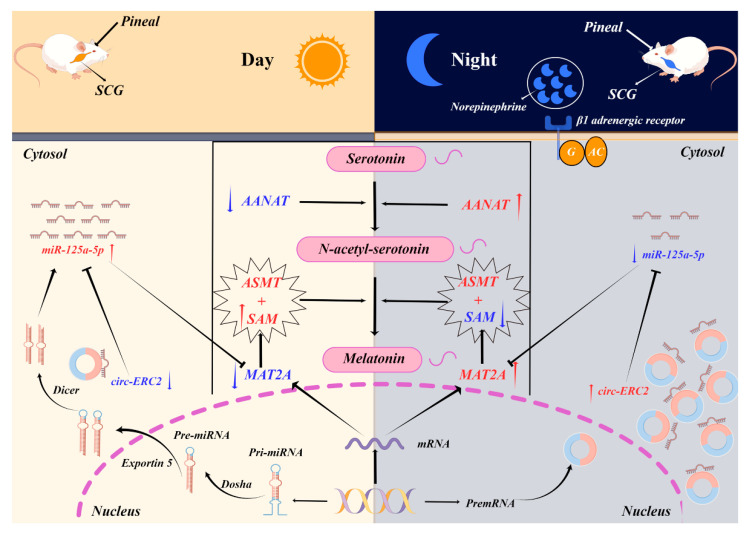
Circ-ERC2 is involved in melatonin synthesis through the miR-125a-5p/MAT2A axis. This figure is drawn using Figdraw (https://www.figdraw.com/static/index.html#/, accessed on 10 September 2022). The unique authorization code is IUWYA559da.

## Data Availability

Not applicable.

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
