# Peer review of "Circ-ERC2 Is Involved in Melatonin Synthesis by Regulating the miR-125a-5p/MAT2A Axis"

_ijms, 2022, doi:10.3390/ijms232415477_

Round 1
Reviewer 1 Report
The article 'Circ-ERC2 Is Involved in Melatonin Synthesis by Regulating the miR-125a-5p/MAT2A axis' showed that mat2a has a role in melatonin synthesis. Moreover, miRNA-125a-5p regulates the expression of mat2a and, through mat2a, is involved in melatonin synthesis. Bioinformatics analysis showed a circular structure of circ-ERC2, which was confirmed by a linear RNase digestion experiment. Furthermore, transfection of pineal cells with circ-ERC2 siRNA indicated that it is involved in melatonin synthesis. The article is well-written, and the research results support the conclusions. However, I suggest some minor revisions.
For the figures to stand alone, it is necessary to explain the abbreviations used, such as NC and INC, in Figures 2 and 3
Authors should check spelling/writing through lines 501 - 503.
The present study addressed the question of the relationship between circ-ERC2, miR-125a-5p, mat2a, and the synthesis of melatonin, the main circadian hormone in mammals. This study proved that mat2a is involved in melatonin synthesis. Also, circ-ERC2 is involved in melatonin synthesis via miR-125a-5p/mat2a axis.
The topic certainly contributes to this field and provides a link between circ-ERC2, mat2a, and melatonin synthesis that has not been investigated so far.
This research provides a link between miRNA, circRNA, mat2a, and melatonin synthesis. Also, the authors showed that the circRNA inhibitor reduces the expression of mat2a, which directly affects the synthesis of melatonin, while simultaneously increasing the expression of miRNA-125a-5p.
The conclusions are in accordance with the results presented in the manuscript. The authors answered the research question and proved the existence of a connection between circ-ERC2, miR-125a-5p, and mat2a and melatonin synthesis.
Cited references are appropriate.
Authors should correct the spelling mistake in Figure 5, line 224 - it should be circ-ERC2, not circR-ERC2.
Author Response
For Reviewer 1:
The article 'Circ-ERC2 Is Involved in Melatonin Synthesis by Regulating the miR-125a-5p/MAT2A axis' showed that mat2a has a role in melatonin synthesis. Moreover, miRNA-125a-5p regulates the expression of mat2a and, through mat2a, is involved in melatonin synthesis. Bioinformatics analysis showed a circular structure of circ-ERC2, which was confirmed by a linear RNase digestion experiment. Furthermore, transfection of pineal cells with circ-ERC2 siRNA indicated that it is involved in melatonin synthesis. The article is well-written, and the research results support the conclusions. However, I suggest some minor revisions.
- For the figures to stand alone, it is necessary to explain the abbreviations used, such as NC and INC, in Figures 2 and 3
Thank you for your correction. We did not describe it clearly, and we have modified it in the original text.
L 147: NC means Mat2a siRNA-NC.
L 182-183: NC means miR-125a-5p mimic-NC and INC means miR-125a-5p inhibitor-NC.
- Authors should check spelling/writing through lines 501 - 503.
Thank you for your valuable advice. We have corrected the description order in the materials and methods.
L 504-508:
All experiments in this study were repeated at least 3 times. The t test of GraphPad Prism 8 was used to compare the significance of the two groups of data, and the multiple comparison data were analyzed for significance using SPSS 19.0 one-way ANOVA. All data are the mean ± standard deviation of three independent biological replicates. P<0.05 was considered to indicate statistical significance.
The present study addressed the question of the relationship between circ-ERC2, miR-125a-5p, mat2a, and the synthesis of melatonin, the main circadian hormone in mammals. This study proved that mat2a is involved in melatonin synthesis. Also, circ-ERC2 is involved in melatonin synthesis via miR-125a-5p/mat2a axis.
The topic certainly contributes to this field and provides a link between circ-ERC2, mat2a, and melatonin synthesis that has not been investigated so far.
This research provides a link between miRNA, circRNA, mat2a, and melatonin synthesis. Also, the authors showed that the circRNA inhibitor reduces the expression of mat2a, which directly affects the synthesis of melatonin, while simultaneously increasing the expression of miRNA-125a-5p.
The conclusions are in accordance with the results presented in the manuscript. The authors answered the research question and proved the existence of a connection between circ-ERC2, miR-125a-5p, and mat2a and melatonin synthesis.
Cited references are appropriate.
- Authors should correct the spelling mistake in Figure 5, line 224 - it should be circ-ERC2, not circR-ERC2.
Thank you for your correction. We have modified it in the original text.
L 225: Figure 5. circ-ERC2 acts as a molecular sponge for miR-125a-5p in pineal cells.
Reviewer 2 Report
This paper presents very interesting and novel data elucidating post-transcription agendas of melatonin biosynthesis and showing that circ-ERC2 can act as a molecular sponge of miR-125a-5p fine-tuning melatonin production in pineal gland by targeting mat2a.
The paper is well written; appropriate statistical approaches were applied.
Paper is expected to be interesting to many readers, however, several important issues deserve clarification before paper can be accepted for publication:
-
Figure 1C, depicted daytime serum melatonin levels are too high for daytime, and the difference between day and night is modest. How can it be explained?
-
Taken high daytime melatonin levels, the next question arises: how sensitive are ELISA Melatonin kits (Enzyme-linked Biotechnology Co., Ltd.) used to determine melatonin levels. What is functional sensitivity and standard curve range of this kit, particularly during daytime, how low is the threshold?
-
At which circadian time (hours after lights-off) pineal glands were extracted, was this time always the same? How much time exactly in hours passed (what was the time lag) between estimated melatonin and sacrifice of the rats? it is indicated at line 393 that 4 days, but was time passed always the same?
-
At what light (L) : dark (D) regimen (hours ratio) rats were kept in the lab before they were sacrificed, what was light intensity and L and D phases?
-
p.5, Line 130. The authors refer to knock-down of mat2a expression in pineal cells. Do they mean interference by siRNA-1? I wonder whether “knockdown” is a proper term in this context. Also, was time factor always kept in-sync among all samples? This procedure is not clearly explained in the Methods section.
-
In the previous publications miR-125a was reported to be involved in the numerous circadian and clock-related processes in the brain (doi: 10.1152/japplphysiol.00940.2012; doi: 10.1016/j.neulet.2007.06.005) and, importantly, may target clock genes (e.g., doi: 10.3390/clockssleep2030022). Notably, miR-125a was reported to be expressed mainly at the end of a dark period in the rat brain (doi: 10.1152/japplphysiol.00940.2012), that is different from what is depicted at Figure 6 of the present manuscript. I invite the authors to provide discussion on this topic.
Round 2
Reviewer 2 Report
The authors correctly addressed main issues. I would now recommend it for publication. However, since ELISA sensitivity reported by authors is above 9 pg/mL, that is apparently too high (cf. https://doi.org/10.1111/jpi.12657), I would recommend to add this sensitivity issue to study limitations.
Author Response
For Reviewer 2:
The authors correctly addressed main issues. I would now recommend it for publication. However, since ELISA sensitivity reported by authors is above 9 pg/mL, that is apparently too high (cf. https://doi.org/10.1111/jpi.12657), I would recommend to add this sensitivity issue to study limitations.
Thank you for your valuable advice. We have modified it in the original text.
L 434-436:
For Melatonin ELISA kit,the sensitivity of ELISA kit is 9.38pg/ml, and the detection range is 15.63-1000pg/ml. It can detect the expression level of melatonin in rats in the samples, and there is no obvious cross-reaction with other analogues.
L 285-286:
However, due to the high sensitivity of ELISA, the use of ELISA to detect the changes of hormone expression in rat serum has certain limitations.